# Profiling 25 Bone Marrow microRNAs in Acute Leukemias and Secondary Nonleukemic Hematopoietic Conditions

**DOI:** 10.3390/biomedicines8120607

**Published:** 2020-12-14

**Authors:** Igor B. Kovynev, Sergei E. Titov, Pavel S. Ruzankin, Mechti M. Agakishiev, Yuliya A. Veryaskina, Viktor M. Nedel’ko, Tatiana I. Pospelova, Igor F. Zhimulev

**Affiliations:** 1Department of Therapy, Hematology and Transfusiology, Novosibirsk State Medical University, 630091 Novosibirsk, Russia; kovin_gem@mail.ru (I.B.K.); m_agakishiev@mail.ru (M.M.A.); depart04@mail.ru (T.I.P.); 2Laboratory of Molecular Genetics, Department of the Structure and Function of Chromosomes, Institute of Molecular and Cellular Biology, Siberian Branch of the Russian Academy of Sciences, 630090 Novosibirsk, Russia; titovse78@gmail.com (S.E.T.); zhimulev@mcb.nsc.ru (I.F.Z.); 3AO Vector-Best, 630117 Novosibirsk, Russia; 4Sobolev Institute of Mathematics, Siberian Branch of Russian Academy of Sciences, 630090 Novosibirsk, Russia; ruzankin@math.nsc.ru (P.S.R.); nedelko@math.nsc.ru (V.M.N.); 5Department of Mathematics and Mechanics, Novosibirsk State University, 630090 Novosibirsk, Russia; 6Department of Hematology, City Clinical Hospital #2, 630051 Novosibirsk, Russia; 7Laboratory of Gene Engineering, Institute of Cytology and Genetics, Siberian Branch of the Russian Academy of Sciences, 630090 Novosibirsk, Russia

**Keywords:** acute myeloblastic leukemia, acute lymphoblastic leukemia, microRNA

## Abstract

Introduction: The standard treatment of acute leukemias (AL) is becoming more efficacious and more selective toward the mechanisms via which to suppress hematologic cancers. This tendency in hematology imposes additional requirements on the identification of molecular-genetic features of tumor clones. MicroRNA (miRNA, miR) expression levels correlate with cytogenetic and molecular subtypes of acute leukemias recognized by classification systems. The aim of this work is analyzing the miRNA expression profiles in acute myeloblastic leukemia (AML) and acute lymphoblastic leukemia (ALL) and hematopoietic conditions induced by non-tumor pathologies (NTP). Methods: A total of 114 cytological samples obtained by sternal puncture and aspiration biopsy of bone marrow (22 ALLs, 44 AMLs, and 48 NTPs) were analyzed by real-time PCR regarding preselected 25 miRNAs. For the classification of the samples, logistic regression was used with balancing of comparison group weights. Results: Our results indicated potential feasibility of (i) differentiating ALL+AML from a nontumor hematopoietic pathology with 93% sensitivity and 92% specificity using miR-150:miR-21, miR-20a:miR-221, and miR-24:nf3 (where nf3 is a normalization factor calculated from threshold cycle values of miR-103a, miR-191, and miR-378); (ii) diagnosing ALL with 81% sensitivity and 81% specificity using miR-181b:miR-100, miR-223:miR-124, and miR-24:nf3; and (iii) diagnosing AML with 81% sensitivity and 84% specificity using miR-150:miR-221, miR-100:miR-24, and miR-181a:miR-191. Conclusion: The results presented herein allow the miRNA expression profile to de used for differentiation between AL and NTP, no matter what AL subtype.

## 1. Introduction

Acute leukemia (AL) is a hematologic cancer arising from early hematopoietic precursors that have undergone malignant transformation and for that reason lost their ability to differentiate into mature blood cells. From a clinical perspective, the progression of this type of hematologic cancer may be evident on a hemogram as pancytopenia due to the withdrawal of normal lineages from bone marrow by tumor blasts, as pronounced tumor toxicity, as secondary lesions to internal organs due to blastic infiltration, and as hematopoietic deficiencies (anemia, hemorrhagic syndrome, and infection-based agranulocytosis). Special epidemiological features of ALs make these pathologies some of the most important types of hematologic cancers that are determinants of treatment efficacy of hematologic cancers [1].

The standard treatment of ALs is becoming more efficacious and more selective toward the mechanisms via which to suppress hematologic cancers. Cells that remain in the BM after chemotherapy are believed to be responsible for relapses. This is why protocols for an early and accurate detection of the residual leukemic cells (minimal residual disease (MRD)) are important. MRD monitoring may have importance for clinical decision making, as it allows survival rates and relapse risk to be accurately assessed [2].

Classical cell-based methods used to ascertain relapsed AL, including flow immunocytofluorometry, are well proven and reliable. However, these methods are no aids to monitor early molecular-genetic events in the genomes of the pre-cancerous precursors that have implications for AL progression. This is why the only privilege that a physician armed with standard laboratory methods for detecting a relapse has at the moment is to observe what clonal selection has done rather than to have full control of the process.

Residual leukemic cells are not seldom present in very low numbers, and so their detection requires more sensitive methods [3]. Real-time PCR and droplet digital PCR are some of them. However, MRD detection can be complicated by a phenomenon known as ‘clonal evolution’. Thus, analysis of all molecular markers at diagnosis and first relapse may show that the predominant clone is not always of one and the same origin [4]. There is little doubt that the use of any MRD detection method requires a biomarker or a combination of biomarkers that can clearly differentiate between normal and cancer cells. This is especially difficult in AL due to a high genetic instability of tumor cells’ genome and, as a consequence, a substantial clonal heterogeneity of this disease.

On the other hand, patients who are neither genetically nor epigenetically predisposed to tumor progression/relapse have to rely on standard chemotherapeutic relapse prevention protocols and are exposed to undue risk of side effects and even death.

Analysis of molecular-genetic factors in the pathogenesis of leukemias has allowed to discover a new regulatory mechanism underlying an abnormal function of the key genes responsible for differentiation into myeloid and lymphoid cell types. This mechanism involves microRNAs (miRNAs, miRs), i.e., short noncoding RNAs exerting regulatory action on the expression of target genes, both transcriptionally and translationally [5].

It has been demonstrated that miRNA expression levels correlate with cytogenetic and molecular AL subtypes recognized by classification systems and determine many properties of tumor blasts [5,6]. Aberrations in miRNA expression profiles in AL have been demonstrated [7,8,9]. Today, there is little doubt that miRNAs correlate strongly with the efficacy of standard chemotherapy for AL and clinical outcomes of hematologic cancers [10].

The aim of this work is analyzing the miRNA expression profiles in acute myeloblastic leukemia (AML), acute lymphoblastic leukemia (ALL), and hematopoietic conditions induced by non-tumor pathologies (NTP). Data obtained will allow us to identify the miRNAs that can be used as high-sensitivity biomarkers to detect MRD with.

## 2. Experimental Section

Clinical material. A total of 114 cytological samples were obtained by sternal puncture and aspiration biopsy of bone marrow on the posterior iliac spine. All the cases were AL patients at the Novosibirsk Municipal Hematological Center before treatment initiation. Cytological material was obtained in compliance with Russian laws and regulations, written informed consent was obtained from each patient, and all the data were depersonalized. The study protocol No.15 of May 25, 2020 was approved by the Ethics Committee of Novosibirsk State Medical University.

The types of hematologic cancers included in the study population were ALL (22 specimens) and AML (44 specimens). The characteristics of the groups are shown in Appendix A. Work with healthy bone marrow donors for allogeneic transplantation is beyond the competence of our clinic. Taken together, we decided to go with a control group composed of people who had no hematologic cancer, but had indications for bone marrow examination to exclude one. They were people with secondary anemic and cytopenic conditions, in whom leukemias were not confirmed by myelography (NTP (48 specimens)). The characteristics of the NTP group are shown in Table 1.

Selecting miRNAs. MiRNAs were chosen based on literature data. The experimental analysis involved 25 miRNAs: miR-100-5p, -124-3p, -126-3p, -128-3p, -146a-5p, -150-5p, -155-5p, -18a-5p, -181a-5p, -181b-5p, -196b-5p, -20a-5p, -21-5p, -210-3p, -221-3p, -223-3p, -24-3p, -26a-5p, -29b-3p, -451a, -9-5p, -92a-3p, -96-5p, -99a-5p, and let-7a [10,11,12,13,14,15]. Reference miRNAs were miR-378-3p, -191-5p, and -103a-3p, which were selected by means of our original data and literature data [16]. In some classification variables, the geometric mean of threshold cycle (Ct) values of the three reference miRNAs was employed for normalization as proposed by Vandesompele [17].

Total nucleic acid isolation. Nucleic acid was isolated and, as described by Titov et al., a dried cytological smear was washed into a microcentrifuge tube with three 200 μL portions of guanidine lysis buffer [18]. The sample was vigorously mixed and incubated in a thermal shaker for 15 min at 65 °C. Next, an equal volume of isopropanol was added. The reaction solution was thoroughly mixed and kept at room temperature for 5 min. After centrifugation for 10 min at 14000 *g*, the supernatant was discarded, and the pellet was washed with 500 μL of 70% ethanol and 300 μL of acetone. Finally, the RNA was dissolved in 200 μL of deionized water. If not analyzed immediately, RNA samples were stored at 20 °C.

Oligonucleotide primers and probes. All the oligonucleotides, including fluorescently labeled ones, were synthesized by AO Vector-Best (Novosibirsk, Russia). The oligonucleotides were chosen using an online tool, PrimerQuest (https://eu.idtdna.com/). For each miRNA, several sets of oligonucleotides were chosen, from which those with the highest real-time PCR efficiency were selected. PCR efficiency was assessed by constructing a standard curve for serial dilutions of synthetic miRNA analogs (OOO Biosan, Novosibirsk, Russia) of known concentration. Depending on the system, the E value varied from 91.5% to 99.8%. The sequences of the oligonucleotides are given in Appendix A.

Detection of miRNA by real-time PCR. Mature miRNAs were detected via the method proposed by Chen et al. [19]. For each miRNA, reverse transcription was carried out, followed by real-time PCR as described by Titov et al. [18]. Reverse transcription and PCR for each sample involved one replicate each. Concentrations of some miRNAs were normalized to “nf3,” which is the geometric mean of Ct values of the three reference miRNAs (miR-103a-3p, miR-191-5p, and miR-378-3p), by the 2^−ΔCt^ method [20]. Concentrations of some other miRNAs were normalized to another miRNA (instead of nf3), as described below. In other words, nf3 served as the normalization factor for some classification variables.

Classifying samples. To investigate the association between miRNA concentrations and outcomes in NTP, ALL, and AML, classification variables were created that represent binary logarithms of pairwise ratios of miRNA concentrations; for example, the classification variable miR-150:miR-378 denotes Ct(miR-150) minus Ct(miR-378). The paper by Ivanov et al. explains why it is worthwhile to use ratios corresponding to pairs of markers (where an oncogenic or tumor suppressor miRNA is normalized to another marker miRNA) rather than stand-alone markers (individual miRNAs) normalized to housekeeping genes [21]. Above-mentioned normalization factor nf3 was employed to create some classification variables (i.e., to calculate some ratios), for example, miR-150:nf3 was equal to Ct(miR-150) − Ct(nf3). The reference miRNAs miR-103a-3p, miR-191-5p, and miR-378-3p were included in our analysis as nonreference miRNAs too, even though they are components of nf3. For instance, variables miR-150:miR-378 and miR-191:nf3 were utilized in the classification analysis. A total of 406 of such classification variables based on miRNA Ct values were tested in this analysis.

Primary analysis. For each classification variable, the following comparisons were made: NTP vs. others, ALL vs. others, AML vs. others, and ALL vs. AML. The comparisons were carried out by the exact Mann–Whitney test. In accordance with the Bonferroni approach to multiple comparisons, differences with *p*-values less than 0.05/(4 × 406) were considered statistically significant. In addition to the *p*-values, we calculated the following prediction accuracy measures for leave-one-out cross-validation: accuracy, sensitivity, specificity, and receiver-operating characteristic (ROC) area under the curve (AUC) with DeLong’s confidence interval. Predicted values were computed via logistic regression with balancing of the comparison groups’ weights. The accuracy, sensitivity, specificity, and ROC AUC values in the leave-one-out cross-validation were calculated for the threshold value of 0.5.

Secondary analysis. To assess the possibility of improving prediction accuracy, we examined all logistic regression models based on two or three of the created variables. In total, for each of the four comparisons, (406 × 405/2) + (406 × 405 × 404/6) = 11,153,835 models with two or three regressors were tested. Additionally, we tried the following machine learning methods: support vector machine, linear discriminant analysis, and boosting, but their performance was not better than that of logistic regression, and therefore their results are omitted here.

The computations were made in the R software, v.3.6.3 (R Core Team).

## 3. Results

### 3.1. Comparing miRNA Concentrations among ALL, AML, and NTP Samples

Relative concentrations of miRNAs in different sample types were determined by RT-PCR (Figure 1). As an example, the scatterplot of miR-150:nf3 and miR-221:nf3 for individual patients is displayed in Figure 2.

### 3.2. Sample Classification

Although significant differences in the concentrations of seven miRNAs were found between either ALL or AML and NTP samples, none of the miRNAs could serve as a single marker (Figure 1). For this reason, when classifying samples, we used ratios of concentrations for pairs of miRNAs (some of these ratios involved reference miRNAs and/or nf3) in univariate classification models or combinations of such ratios in multivariable classification models.

The results of the comparison between NTP and the others are given in Table 2; those for ALL vs. others, in Table 3; the results on AML vs. others in Table 4, and the comparison between ALL and AML is presented in Table 5. The tables list univariate models for the variables for which *p* × 4 × 406 < 0.05 and those classification variables that are based on normalization to nf3. For the sake of convenience, in the tables, the *p*-values are multiplied by the number of comparisons 4 × 406 to allow the reader to compare the “normalized” *p*-values with the common threshold of 0.05, which is equivalent to comparing the “raw” *p*-values with 0.05/(4 × 406). Additionally, some logistic regression models based on two or three variables with the best cross-validation accuracy are presented in the tables.

According to Table 2, which shows the results of the comparison between NTP and the others, the best prediction accuracy (93.8% sensitivity with 92.4% specificity) was manifested by the classification model based on covariates miR-150:miR-21, miR-20a:miR-221, and miR-24:nf3. Among univariate models, miR-150:miR-378 had the best prediction accuracy (87.5% sensitivity with 72.7% specificity).

According to Table 3, which presents the results of the comparison between ALL and the others, the model based on covariates miR-181b:miR-100, miR-223:miR-124, and miR-24:nf3 had the best prediction accuracy (81.8% sensitivity with 81.5% specificity).

According to Table 4, which lists findings of the comparison between AML and the others, the best prediction accuracy (81.8% sensitivity with 84.3% specificity) was shown the model based on the covariates miR-150:miR-221, miR-100:miR-24, and miR-181a:miR-191. Among univariate models, miR-150:miR-191 had the best prediction accuracy (68.2% sensitivity with 78.6% specificity).

According to Table 5, which lists the results for the comparison between ALL and AML, the best prediction accuracy (86.4% sensitivity with 84.1% specificity) belonged to the classification model based on covariates miR-100:miR-124, miR-24:miR-26a, and miR-24:miR-9, the model based on covariates miR-100:miR-124, miR-24:miR-26a, and miR-26a:miR-9, and the model based on mirR-100:miR-124, miR-24:miR-9, and miR-26a:miR-9.

## 4. Discussion

The expression of miRNA is always different between tumors and healthy tissues or a secondary nontumor pathology. Additionally, differences in the miRNA profiles among different tumor types and among different stages of the same malignant tumor are not uncommon [22,23]. Because miRNAs are highly stable in tissues and body fluids, they appear to be promising diagnostic markers.

In this work, we compared the expression profiles of 25 miRNA in bone marrow samples from new cases of AML, ALL, and NTPs.

Our study indicates that each of the aforementioned hematopoietic bone marrow disorders may be identified through profiling of miRNAs. The results of our analysis show that some of our classification variables are statistically significantly associated with the comparison groups.

The results from our multivariable models, even though they may be subject to overfitting to some extent, cannot be explained by overfitting alone. For example, in the comparison of NTP with ALL+AML, the best accuracy (93.8% sensitivity with 92.4% specificity) was shown by the classification model based on covariates miR-150:miR-21, miR-20a:miR-221, and miR-24:nf3. In the exact Mann–Whitney test comparing the values predicted by logistic regression during the leave-one-out cross-validation between the comparison groups, we obtained a *p*-value < 2.2 × 10^−16^. On the other hand, dividing 0.05 by the total number of comparisons yields 0.05/(4 × (11,153,835 + 406)) = 1.1 × 10^−9^, which is at least 5 × 10^6^ times greater than the above *p*-value.

As for individual miRNAs, NTP was best discriminated from ALL+AML by miR-150 (85.4% sensitivity with 71.2% specificity for the corresponding variable, miR-150:nf3), ALL was best discriminated from AML+NTP by miRNA-223 (59.1% sensitivity with 70.7% sensitivity for the corresponding variable, miR-223:nf3), AML was best differentiated from ALL+NTP by miRNA-150 (63.6% sensitivity with 80.0% specificity for miR-150:nf3), and, finally, ALL was best differentiated from AML by miRNA-100 (63.6% sensitivity with 65.9% specificity for miR-100:nf3), although expression deregulation of these miRNAs has been described [14]. AML is a special case because changes in the expression of miRNA-126, -29b, and -26a are reported more often [24,25,26,27] in comparison with miRNA-150. In ALL, changes in the expression of miRNA-150 and -155 are detected more often [28,29,30,31,32] as compared to miRNA-223.

Inconsistency between miRNA profiling results of different studies is a common problem. This could be because different authors use dissimilar control groups; this is especially true for the research on hematologic cancers. Another source of inconsistency is a relatively large error in measurements of miRNA concentrations whether by RT-PCR or via microarray technologies. Because of the measurement errors, potentially significant under-three-fold changes in miRNA concentrations may not be taken into account accurately. Thus, in this study, the most reliable results, which at the same time were most consistent between different studies, were obtained only for miRNAs whose cancer-specific differences in concentration were the greatest.

Therefore, our results point to potential feasibility of (i) discriminating ALL+AML from NTP with 93% sensitivity of and 92% specificity using miR-150:miR-21, miR-20a:miR-221, and miR-24:nf3; (ii) diagnosing ALL with 80% sensitivity and 81% specificity by means of miR-196b:miR-221, miR-223:miR-378, and miR-100:miR-29b; and (iii) diagnosing AML with 80% sensitivity and 83% specificity with the help of miR-150:miR-100, miR-181a:miR-191, and miR-221:miR-26a.

The results presented herein allow the miRNA expression profile to be used for differentiation between AL and NTP, no matter what AL subtype.

Effective treatment for acute leukemias strongly depends on our understanding of the basics of their genesis. One of the fundamental mechanisms underlying leukemias is the phenomenon known as ‘clonal selection’. This pathogenetic mechanism is in fact responsible for all the clinical features of tumor relapse to come. One of the problems where the data obtained can be helpful is the detection of MRD. There is little doubt that the use of any MRD detection method requires a biomarker or a combination of biomarkers that can clearly differentiate between normal and cancer cells. Data obtained allow miRNAs to be seen as promising biomarkers with sufficient sensitivity and specificity to detect MRD, no matter what clonal nature of the disease.

The aim of this publication is to present the results of the first, pilot stage of the research project. With reliance on a wide spectrum of miRNAs, we have confirmed the feasibility of using analyses of miRNA patterns for solving differential problems when analyzing bone marrow samples.

To this end, it is necessary to do testing with larger sample sizes. The obvious extension of this work will be to analyze miRNA expression profiles after consolidation chemotherapy and to see whether they correlate with the patient’s clinical data.

## Figures and Tables

**Figure 1 biomedicines-08-00607-f001:**
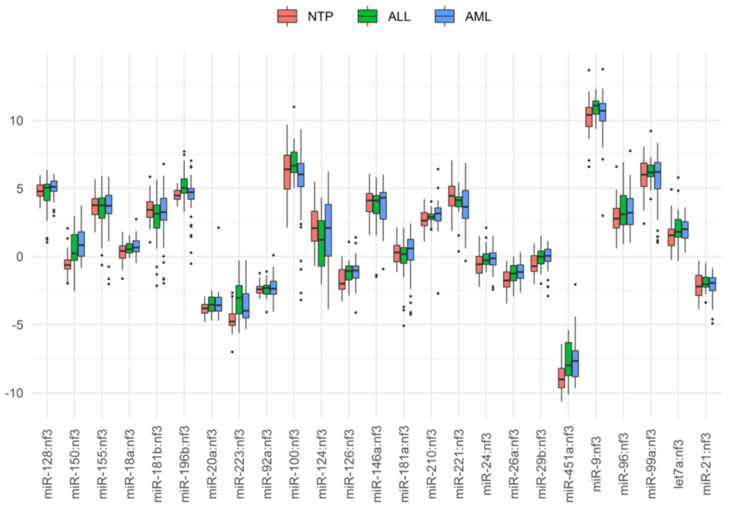
Boxplot for binary logarithms of some miRNA concentrations, those that were normalized to nf3. The boxes depict medians with the 1st and 3rd quartiles.

**Figure 2 biomedicines-08-00607-f002:**
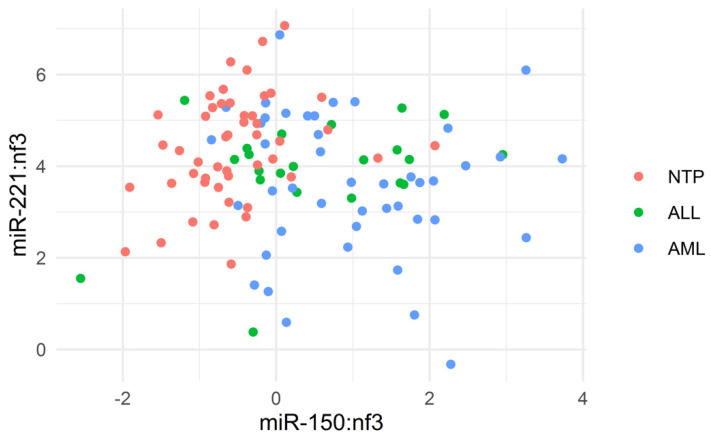
The scatterplot of miR-150:nf3 and miR-221:nf3 for individual patients.

**Table 1 biomedicines-08-00607-t001:** Clinical data of non-cancerous blood diseases (NTP) (*n* = 48).

Characteristic	*n* (%)
**Gender**	
Male	23 (48)
Female	25 (52)
**Age**	
>60 years	11 (23)
<60 years	37 (77)
Median hemoglobin, g/L	90
Median WBC count, ×109/L	6.7
Median ANC, /dL	5
Median platelet count, ×109/L	200.5
**Sybtype**	
Iron-deficiency anemia	28 (58)
hemolytic anemia	3 (6)
B12 deficiency anemia	5 (10)
chronic disease anemia	6 (13)
immune thrombocytopenia	5 (10)
aplastic anemia	1 (2)

Abbreviations: WBC = white blood cell; ANC = absolute neutrophil count.

**Table 2 biomedicines-08-00607-t002:** Comparison of NTP vs. others.

	*p*-Value × 4 × 406	CV Accuracy	CV Sensitivity	CV Specificity	CV AUC
miR-150:miR-21 + miR-20a:miR-221 + miR-24:nf3		0.930	0.938	0.924	0.949 (0.910, 0.989)
miR-150:miR-223 + miR-150:miR-221 + miR-126:miR-191		0.921	0.917	0.924	0.950 (0.910, 0.991)
miR-150:miR-223 + miR-150:nf3 + miR-126:miR-221		0.921	0.917	0.924	0.959 (0.926, 0.993)
miR-150:miR-223 + miR-223:miR-221 + miR-126:miR-191		0.921	0.917	0.924	0.950 (0.910, 0.991)
miR-150:miR-223 + miR-223:nf3 + miR-126:miR-221		0.921	0.917	0.924	0.959 (0.926, 0.993)
miR-150:miR-221 + miR-223:miR-221 + miR-126:miR-191		0.921	0.917	0.924	0.950 (0.910, 0.991)
miR-150:nf3 + miR-20a:miR-221 + miR-24:miR-103a		0.921	0.917	0.924	0.946 (0.900, 0.991)
miR-150:nf3 + miR-223:nf3 + miR-126:miR-221		0.921	0.917	0.924	0.959 (0.926, 0.993)
miR-150:miR-146a + miR-155:miR-221 + miR-24:miR-378		0.921	0.938	0.909	0.951 (0.910, 0.993)
miR-150:miR-221 + miR-196b:miR-99a + miR-24:nf3		0.921	0.958	0.894	0.950 (0.909, 0.992)
miR-223:miR-378 + miR-221:miR-24 + miR-29b:nf3		0.921	0.958	0.894	0.943 (0.895, 0.990)
miR-223:nf3 + miR-221:miR-24		0.886	0.917	0.864	0.919 (0.865, 0.972)
miR-150:miR-221 + miR-24:miR-378		0.877	0.917	0.848	0.931 (0.886, 0.976)
miR-223:miR-221 + miR-126:miR-103a		0.877	0.875	0.879	0.929 (0.881, 0.977)
miR-223:miR-221 + miR-126:miR-191		0.877	0.896	0.864	0.932 (0.888, 0.975)
miR-223:miR-221 + miR-29b:nf3		0.877	0.875	0.879	0.930 (0.880, 0.980)
miR-150:miR-378	0.00000000071	0.789	0.875	0.727	0.863 (0.794, 0.932)
miR-150:nf3	0.0000000024	0.772	0.854	0.712	0.857 (0.785, 0.929)
miR-221:miR-24	0.000000044	0.746	0.792	0.712	0.836 (0.764, 0.908)
miR-223:miR-221	0.00000056	0.719	0.792	0.667	0.820 (0.744, 0.895)
miR-150:miR-221	0.00000099	0.728	0.792	0.682	0.817 (0.741, 0.893)
miR-150:miR-191	0.0000021	0.746	0.812	0.697	0.811 (0.730, 0.892)
miR-223:miR-378	0.0000045	0.693	0.792	0.621	0.806 (0.726, 0.886)
miR-150:miR-92a	0.000043	0.746	0.792	0.712	0.788 (0.703, 0.873)
miR-150:miR-103a	0.000045	0.719	0.771	0.682	0.785 (0.699, 0.871)
miR-128:miR-150	0.000068	0.693	0.729	0.667	0.782 (0.698, 0.867)
miR-150:miR-146a	0.00011	0.719	0.792	0.667	0.779 (0.695, 0.863)
miR-150:miR-181a	0.00011	0.711	0.792	0.652	0.778 (0.695, 0.861)
miR-451a:miR-103a	0.00015	0.684	0.667	0.697	0.775 (0.690, 0.860)
miR-150:miR-181b	0.00017	0.711	0.812	0.636	0.773 (0.687, 0.860)
miR-92a:miR-451a	0.00032	0.667	0.688	0.652	0.769 (0.682, 0.855)
miR-221:miR-26a	0.00045	0.728	0.771	0.697	0.764 (0.675, 0.853)
miR-126:miR-221	0.00059	0.728	0.750	0.712	0.764 (0.676, 0.852)
miR-150:miR-21	0.00099	0.719	0.729	0.712	0.759 (0.668, 0.849)
miR-451a:nf3	0.0015	0.649	0.667	0.636	0.753 (0.665, 0.842)
miR-451a:miR-21	0.0016	0.667	0.792	0.576	0.753 (0.662, 0.844)
miR-181b:miR-223	0.0019	0.640	0.833	0.500	0.751 (0.662, 0.841)
miR-221:miR-451a	0.0020	0.693	0.750	0.652	0.751 (0.663, 0.840)
miR-451a:miR-378	0.0032	0.667	0.750	0.606	0.745 (0.656, 0.834)
miR-221:miR-9	0.0034	0.702	0.771	0.652	0.747 (0.656, 0.837)
miR-221:miR-29b	0.0042	0.711	0.729	0.697	0.747 (0.656, 0.838)
miR-26a:miR-378	0.0050	0.675	0.646	0.697	0.741 (0.649, 0.833)
miR-150:miR-20a	0.0054	0.693	0.688	0.697	0.744 (0.653, 0.835)
miR-29b:miR-378	0.0069	0.711	0.708	0.712	0.740 (0.647, 0.832)
miR-150:miR-18a	0.0074	0.693	0.729	0.667	0.739 (0.647, 0.831)
miR-24:miR-378	0.0091	0.667	0.625	0.697	0.735 (0.638, 0.833)
miR-150:miR-99a	0.011	0.702	0.750	0.667	0.732 (0.639, 0.825)
miR-451a:let7a	0.016	0.684	0.771	0.621	0.729 (0.637, 0.822)
miR-223:nf3	0.017	0.675	0.792	0.591	0.725 (0.633, 0.818)
miR-150:miR-196b	0.017	0.667	0.792	0.576	0.732 (0.638, 0.827)
miR-126:miR-378	0.019	0.675	0.667	0.682	0.729 (0.633, 0.825)
miR-150:miR-155	0.026	0.684	0.792	0.606	0.727 (0.633, 0.821)
miR-20a:miR-451a	0.026	0.632	0.646	0.621	0.724 (0.629, 0.820)
miR-451a:miR-191	0.032	0.649	0.667	0.636	0.722 (0.628, 0.815)
miR-29b:nf3	0.18	0.684	0.646	0.712	0.698 (0.597, 0.799)
miR-126:nf3	0.41	0.649	0.646	0.652	0.686 (0.581, 0.791)
miR-26a:nf3	0.70	0.623	0.604	0.636	0.681 (0.581, 0.781)
miR-20a:nf3	>1	0.588	0.604	0.576	0.636 (0.535, 0.737)
miR-210:nf3	>1	0.632	0.583	0.667	0.628 (0.521, 0.735)
let7a:nf3	>1	0.570	0.625	0.530	0.627 (0.519, 0.735)
miR-221:nf3	>1	0.570	0.604	0.545	0.624 (0.520, 0.728)
miR-24:nf3	>1	0.579	0.542	0.606	0.628 (0.520, 0.735)
miR-196b:nf3	>1	0.614	0.562	0.652	0.621 (0.517, 0.724)
miR-18a:nf3	>1	0.570	0.542	0.591	0.614 (0.508, 0.720)
miR-96:nf3	>1	0.561	0.646	0.500	0.608 (0.504, 0.713)
miR-9:nf3	>1	0.614	0.500	0.697	0.597 (0.489, 0.705)
miR-128:nf3	>1	0.596	0.521	0.652	0.543 (0.433, 0.652)
miR-124:nf3	>1	0.553	0.583	0.530	0.553 (0.448, 0.659)
miR-21:nf3	>1	0.553	0.542	0.561	0.534 (0.420, 0.648)
miR-181b:nf3	>1	0.579	0.667	0.515	0.553 (0.447, 0.659)
miR-92a:nf3	>1	0.544	0.562	0.530	0.490 (0.383, 0.597)
miR-155:nf3	>1	0.491	0.583	0.424	0.462 (0.356, 0.569)
miR-100:nf3	>1	0.456	0.521	0.409	0.500 (0.389, 0.611)
miR-181a:nf3	>1	0.465	0.562	0.394	0.523 (0.416, 0.630)
miR-99a:nf3	>1	0.439	0.521	0.379	0.591 (0.486, 0.696)
miR-146a:nf3	>1	0.465	0.604	0.364	0.511 (0.404, 0.618)

In this table, sensitivity is the proportion of NTP patients that are correctly identified as such, and specificity is the proportion of ALL+AML patients correctly identified as such.

**Table 3 biomedicines-08-00607-t003:** Comparison of ALL vs. others.

	*p*-Value × 4 × 406	CV Accuracy	CV Sensitivity	CV Specificity	CV AUC
miR-181b:miR-100 + miR-223:miR-124 + miR-24:nf3		0.816	0.818	0.815	0.796 (0.679, 0.914)
miR-155:miR-124 + miR-181b:miR-100 + miR-223:miR-103a		0.807	0.818	0.804	0.839 (0.736, 0.941)
miR-155:miR-378 + miR-181b:miR-223 + miR-100:miR-210		0.807	0.818	0.804	0.829 (0.732, 0.926)
miR-196b:miR-124 + miR-92a:miR-24 + miR-100:miR-181a		0.807	0.818	0.804	0.850 (0.763, 0.936)
miR-155:miR-378 + miR-181b:miR-196b + miR-223:miR-146a		0.807	0.773	0.815	0.768 (0.646, 0.890)
miR-155:miR-378 + miR-181b:miR-223 + miR-196b:miR-24		0.807	0.773	0.815	0.772 (0.663, 0.882)
miR-155:miR-100 + miR-196b:miR-124 + miR-223:miR-92a		0.807	0.727	0.826	0.801 (0.689, 0.914)
miR-181b:miR-223 + miR-196b:miR-103a + miR-100:miR-124		0.807	0.727	0.826	0.796 (0.674, 0.918)
miR-196b:miR-124 + miR-223:miR-26a + miR-99a:miR-378		0.807	0.727	0.826	0.792 (0.673, 0.911)
miR-196b:nf3 + miR-223:miR-181a + miR-126:miR-210		0.807	0.727	0.826	0.726 (0.590, 0.862)
miR-223:miR-26a + miR-100:miR-181a + miR-451a:miR-103a		0.807	0.727	0.826	0.777 (0.658, 0.896)
miR-196b:miR-181a + miR-92a:miR-24 + miR-221:miR-21		0.807	0.682	0.837	0.719 (0.584, 0.855)
miR-155:miR-92a + miR-181b:miR-196b + miR-221:miR-451a		0.807	0.545	0.870	0.698 (0.567, 0.829)
miR-196b:miR-181a + miR-20a:miR-451a + let7a:miR-21		0.807	0.545	0.870	0.643 (0.488, 0.798)
miR-155:miR-92a + miR-181b:miR-126 + miR-196b:miR-181a		0.807	0.500	0.880	0.650 (0.496, 0.803)
miR-181b:miR-223 + miR-196b:miR-103a		0.789	0.682	0.815	0.727 (0.596, 0.858)
miR-18a:miR-451a + miR-181a:miR-24		0.789	0.545	0.848	0.647 (0.502, 0.793)
miR-181b:miR-223 + miR-221:miR-9		0.789	0.545	0.848	0.727 (0.616, 0.838)
miR-128:miR-21 + miR-223:miR-181a		0.789	0.500	0.859	0.617 (0.452, 0.781)
miR-196b:nf3	>1	0.649	0.591	0.663	0.703 (0.570, 0.836)
miR-223:nf3	>1	0.684	0.591	0.707	0.686 (0.548, 0.824)
miR-100:nf3	>1	0.596	0.591	0.598	0.643 (0.527, 0.759)
miR-9:nf3	>1	0.570	0.682	0.543	0.625 (0.502, 0.748)
miR-451a:nf3	>1	0.596	0.500	0.620	0.595 (0.453, 0.737)
miR-124:nf3	>1	0.588	0.545	0.598	0.595 (0.459, 0.732)
miR-150:nf3	>1	0.614	0.455	0.652	0.570 (0.439, 0.700)
miR-29b:nf3	>1	0.526	0.591	0.511	0.583 (0.450, 0.715)
miR-126:nf3	>1	0.570	0.636	0.554	0.564 (0.436, 0.691)
miR-24:nf3	>1	0.535	0.455	0.554	0.574 (0.449, 0.699)
miR-21:nf3	>1	0.518	0.500	0.522	0.564 (0.439, 0.689)
miR-181a:nf3	>1	0.588	0.364	0.641	0.558 (0.428, 0.688)
miR-181b:nf3	>1	0.623	0.409	0.674	0.558 (0.409, 0.707)
miR-20a:nf3	>1	0.596	0.545	0.609	0.509 (0.361, 0.658)
let7a:nf3	>1	0.570	0.455	0.598	0.549 (0.409, 0.689)
miR-96:nf3	>1	0.579	0.455	0.609	0.535 (0.384, 0.686)
miR-26a:nf3	>1	0.518	0.545	0.511	0.506 (0.372, 0.641)
miR-146a:nf3	>1	0.596	0.364	0.652	0.502 (0.366, 0.637)
miR-99a:nf3	>1	0.500	0.636	0.467	0.523 (0.408, 0.637)
miR-18a:nf3	>1	0.491	0.455	0.500	0.502 (0.379, 0.625)
miR-155:nf3	>1	0.561	0.364	0.609	0.499 (0.348, 0.650)
miR-128:nf3	>1	0.588	0.364	0.641	0.489 (0.334, 0.644)
miR-221:nf3	>1	0.500	0.318	0.543	0.602 (0.482, 0.722)
miR-210:nf3	>1	0.482	0.500	0.478	0.650 (0.536, 0.763)
miR-92a:nf3	>1	0.447	0.364	0.467	0.629 (0.507, 0.752)

In this table, sensitivity is the proportion of ALL patients that are correctly identified as such, and specificity is the proportion of NTP+AML patients correctly identified as such.

**Table 4 biomedicines-08-00607-t004:** Comparison of AML vs. others.

	*p*-Value × 4 × 406	CV Accuracy	CV Sensitivity	CV Specificity	CV AUC
miR-150:miR-221 + miR-100:miR-24 + miR-181a:miR-191		0.833	0.818	0.843	0.868 (0.803, 0.934)
miR-150:miR-221 + miR-100:miR-124 + miR-26a:nf3		0.825	0.841	0.814	0.882 (0.821, 0.944)
miR-150:miR-100 + miR-181a:miR-221 + miR-24:nf3		0.825	0.818	0.829	0.872 (0.807, 0.937)
miR-150:miR-100 + miR-181a:nf3 + miR-221:miR-24		0.825	0.818	0.829	0.882 (0.819, 0.944)
miR-150:miR-21 + miR-18a:miR-92a + miR-26a:miR-191		0.825	0.795	0.843	0.831 (0.754, 0.908)
miR-150:nf3 + miR-20a:miR-92a + miR-100:miR-124		0.825	0.795	0.843	0.843 (0.765, 0.921)
miR-223:miR-100 + miR-146a:miR-103a + miR-221:miR-451a		0.825	0.795	0.843	0.836 (0.755, 0.917)
miR-223:miR-103a + miR-100:miR-451a + miR-146a:miR-221		0.825	0.795	0.843	0.823 (0.741, 0.905)
miR-100:miR-126 + miR-146a:miR-221 + miR-26a:miR-21		0.825	0.773	0.857	0.822 (0.736, 0.908)
miR-128:miR-221 + miR-20a:miR-100 + miR-24:nf3		0.825	0.773	0.857	0.812 (0.726, 0.898)
miR-150:miR-221 + miR-196b:miR-24 + miR-100:miR-99a		0.825	0.773	0.857	0.819 (0.730, 0.907)
miR-181b:miR-100 + miR-146a:miR-103a + miR-221:miR-451a		0.825	0.773	0.857	0.834 (0.752, 0.915)
miR-150:miR-221 + miR-26a:miR-103a		0.798	0.795	0.800	0.837 (0.763, 0.911)
miR-150:miR-191 + miR-124:miR-221		0.781	0.773	0.786	0.789 (0.700, 0.878)
miR-150:miR-21 + miR-26a:miR-191		0.781	0.750	0.800	0.823 (0.745, 0.901)
miR-150:miR-191 + miR-210:miR-21		0.781	0.727	0.814	0.803 (0.719, 0.887)
miR-150:miR-191 + miR-181b:miR-124		0.781	0.705	0.829	0.780 (0.689, 0.872)
miR-150:miR-100 + miR-26a:miR-378		0.781	0.682	0.843	0.808 (0.728, 0.889)
miR-150:nf3	0.000022	0.737	0.636	0.800	0.794 (0.712, 0.876)
miR-150:miR-191	0.000057	0.746	0.682	0.786	0.786 (0.699, 0.874)
miR-150:miR-378	0.00016	0.711	0.614	0.771	0.775 (0.690, 0.860)
miR-150:miR-100	0.0011	0.702	0.659	0.729	0.757 (0.665, 0.849)
miR-150:miR-221	0.0015	0.719	0.659	0.757	0.756 (0.659, 0.854)
miR-150:miR-103a	0.0072	0.719	0.682	0.743	0.741 (0.646, 0.837)
miR-150:miR-196b	0.0072	0.684	0.568	0.757	0.741 (0.645, 0.838)
miR-150:miR-21	0.0076	0.684	0.682	0.686	0.739 (0.645, 0.833)
miR-150:miR-92a	0.020	0.711	0.636	0.757	0.729 (0.631, 0.828)
miR-221:miR-24	0.022	0.711	0.636	0.757	0.728 (0.630, 0.825)
miR-26a:miR-21	0.028	0.711	0.636	0.757	0.726 (0.624, 0.829)
miR-221:miR-26a	0.042	0.693	0.636	0.729	0.718 (0.613, 0.822)
miR-100:miR-26a	0.043	0.667	0.659	0.671	0.721 (0.624, 0.819)
miR-451a:nf3	> 1	0.649	0.614	0.671	0.675 (0.575, 0.775)
miR-26a:nf3	> 1	0.596	0.614	0.586	0.649 (0.547, 0.751)
miR-29b:nf3	> 1	0.623	0.705	0.571	0.634 (0.527, 0.740)
miR-126:nf3	> 1	0.570	0.568	0.571	0.630 (0.526, 0.734)
miR-210:nf3	> 1	0.623	0.636	0.614	0.629 (0.520, 0.737)
miR-221:nf3	> 1	0.605	0.568	0.629	0.613 (0.500, 0.725)
miR-128:nf3	> 1	0.544	0.636	0.486	0.604 (0.497, 0.712)
miR-100:nf3	> 1	0.588	0.477	0.657	0.608 (0.501, 0.715)
miR-223:nf3	> 1	0.596	0.477	0.671	0.591 (0.483, 0.699)
miR-18a:nf3	> 1	0.570	0.591	0.557	0.592 (0.483, 0.700)
miR-20a:nf3	> 1	0.570	0.500	0.614	0.584 (0.474, 0.694)
let7a:nf3	> 1	0.570	0.523	0.600	0.573 (0.467, 0.679)
miR-24:nf3	> 1	0.579	0.591	0.571	0.564 (0.456, 0.673)
miR-96:nf3	> 1	0.561	0.477	0.614	0.558 (0.449, 0.666)
miR-181a:nf3	> 1	0.421	0.182	0.571	0.777 (0.685, 0.869)
miR-155:nf3	> 1	0.342	0.227	0.414	0.867 (0.797, 0.936)
miR-9:nf3	> 1	0.500	0.341	0.600	0.572 (0.463, 0.682)
miR-92a:nf3	> 1	0.509	0.477	0.529	0.498 (0.380, 0.616)
miR-146a:nf3	> 1	0.491	0.318	0.600	0.629 (0.518, 0.740)
miR-99a:nf3	> 1	0.544	0.432	0.614	0.505 (0.388, 0.622)
miR-181b:nf3	> 1	0.570	0.455	0.643	0.486 (0.371, 0.602)
miR-21:nf3	> 1	0.465	0.432	0.486	0.586 (0.480, 0.693)
miR-196b:nf3	> 1	0.491	0.386	0.557	0.507 (0.393, 0.621)
miR-124:nf3	> 1	0.377	0.227	0.471	0.827 (0.745, 0.910)

In this table, sensitivity is the proportion of AML patients that are correctly identified as such, and specificity is the proportion of NTP+ALL patients correctly identified as such.

**Table 5 biomedicines-08-00607-t005:** Comparison of ALL vs. AML.

	*p*-Value × 4 × 406	CV Accuracy	CV Sensitivity	CV Specificity	CV AUC
miR-100:miR-124 + miR-24:miR-26a + miR-24:miR-9		0.848	0.864	0.841	0.893 (0.809, 0.976)
miR-100:miR-124 + miR-24:miR-26a + miR-26a:miR-9		0.848	0.864	0.841	0.893 (0.809, 0.976)
miR-100:miR-124 + miR-24:miR-9 + miR-26a:miR-9		0.848	0.864	0.841	0.893 (0.809, 0.976)
miR-155:miR-181b + miR-100:miR-124 + miR-24:miR-26a		0.848	0.818	0.864	0.897 (0.810, 0.984)
miR-155:miR-124 + miR-181b:miR-100 + miR-24:miR-26a		0.848	0.818	0.864	0.893 (0.802, 0.983)
miR-20a:miR-9 + miR-100:miR-124 + miR-24:miR-26a		0.848	0.818	0.864	0.871 (0.781, 0.961)
miR-223:miR-124 + miR-92a:miR-100		0.773	0.773	0.773	0.794 (0.682, 0.907)
miR-100:miR-124 + miR-24:miR-26a		0.773	0.727	0.795	0.851 (0.756, 0.946)
miR-223:miR-124 + miR-100:miR-26a		0.773	0.682	0.818	0.818 (0.715, 0.921)
miR-100:nf3	>1	0.652	0.636	0.659	0.692 (0.561, 0.824)
miR-196b:nf3	>1	0.591	0.591	0.591	0.657 (0.514, 0.800)
miR-181a:nf3	>1	0.576	0.318	0.705	0.560 (0.416, 0.703)
miR-210:nf3	>1	0.621	0.727	0.568	0.572 (0.428, 0.716)
miR-150:nf3	>1	0.530	0.591	0.500	0.568 (0.416, 0.721)
miR-223:nf3	>1	0.652	0.591	0.682	0.565 (0.408, 0.722)
miR-124:nf3	>1	0.530	0.545	0.523	0.544 (0.399, 0.690)
miR-9:nf3	>1	0.530	0.682	0.455	0.569 (0.426, 0.712)
miR-128:nf3	>1	0.606	0.409	0.705	0.568 (0.409, 0.727)
miR-221:nf3	>1	0.606	0.682	0.568	0.544 (0.401, 0.688)
miR-26a:nf3	>1	0.561	0.545	0.568	0.507 (0.351, 0.664)
miR-21:nf3	>1	0.485	0.500	0.477	0.472 (0.325, 0.619)
miR-146a:nf3	>1	0.515	0.227	0.659	0.438 (0.292, 0.584)
miR-181b:nf3	>1	0.606	0.409	0.705	0.496 (0.341, 0.650)
miR-155:nf3	>1	0.545	0.364	0.636	0.505 (0.349, 0.661)
miR-18a:nf3	>1	0.530	0.636	0.477	0.485 (0.333, 0.636)
miR-99a:nf3	>1	0.500	0.636	0.432	0.485 (0.345, 0.624)
miR-92a:nf3	>1	0.470	0.455	0.477	0.548 (0.404, 0.691)
miR-24:nf3	>1	0.455	0.364	0.500	0.518 (0.367, 0.668)
let7a:nf3	>1	0.485	0.409	0.523	0.586 (0.434, 0.738)
miR-29b:nf3	>1	0.424	0.455	0.409	0.689 (0.555, 0.823)
miR-126:nf3	>1	0.455	0.409	0.477	0.753 (0.625, 0.881)
miR-96:nf3	>1	0.515	0.364	0.591	0.644 (0.500, 0.787)
miR-451a:nf3	>1	0.470	0.500	0.455	0.679 (0.540, 0.818)
miR-20a:nf3	>1	0.485	0.591	0.432	0.595 (0.448, 0.742)

In this table, sensitivity is the proportion of ALL patients that are correctly identified as such, and specificity is the proportion of AML patients correctly identified as such.

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
