# Peer review of "Profiling 25 Bone Marrow microRNAs in Acute Leukemias and Secondary Nonleukemic Hematopoietic Conditions"

_biomedicines, 2020, doi:10.3390/biomedicines8120607_

Round 1

Reviewer 1 Report

The authors investigated the expression profile of 25 miRNAs in 114 bone marrow samples including newly diagnosed AML, ALL and  non-tumor pathologies with the aim to discriminate among different diagnoses.

They identified some miRNAs combinations able to differentiate AML and ALL from non—tumor pathologic conditions suggesting their potential use non only in diagnosis, but also in MRD evaluation.

The paper is interesting and underline the role of regulatory molecules in AL pathogenesis and progression, despite the wide variability of miRNAs expression in different diseases.

There are just some point that need clarification:

  • They compared ALL and AML with non—tumor pathologies bone marrow samples. Why did they not employ normal bone marrow? The miRNA expression in healthy people could be different.
  • The analysis was performed at onset of disease. Have the authors any idea of the expression at relapse?
  • Did they identify a value predicting resistance to therapy or relapse probability?
  • They propones miRNAS as biomarkers to detect MRD irrespective to clonal pathway of disease. However, could the miRNA expression be influenced by the new target therapies?

Please add some speculation on this subject

Reviewer 2 Report

Biomedicines review

The authors have a useful and practical goal in this study: they wish to utilize the analysis of a panel of miRNAs to enable detecting minimal residual disease (MRD) in acute leukemia samples. They reason that certain miRNAs are well expressed in certain cell types and thus would be useful to distinguish normal nonmalignant cells from malignant ones. The authors chose miRNAs from the literature. (It would have been good to hear them clarify what sorts of miRNAs these are…what their target genes are).

Having decided what miRNAs they are going to study, the authors then set out to prove that the panel of miRNAs they have selected will be useful since they can be used to distinguish AML from ALL from non malignant hematopoietic conditions. Without reading Suppl table 1, this reviewer continued reading the manuscript till the end, since the Suppl data file was not able to be downloaded by this reviewer. A few days later, the editorial staff sent me the Suppl Data, and I realized that the authors are performing this complicated endeavor to how the difference between iron deficiency anaemia (58% of the cases) (they say in the text that some of the patients had a concurrent GI malignancy but we do not know which of these patients have cancer and which do not. In addition there are chronic disease anemias, B12 deficiency, hemolytic anemia, and then they include 10% cases of ITP, and one case of aplastic anemia.

Overall the idea of this manuscript is reasonable if the authors are looking for a new way to analyze MRD in leukemia. This is always welcome as the authors point out, the characteristics of relapsed cells are not always the same as the original. This miRNA panel theoretically can avoid this pitfall in missing the diagnosis.

However these considerations aside, the two main problems with this manuscript are:

  1. There did not find not particularly impressive rates of distinguishing AML from ALL (around 80% diagnosis of ALL) and, although the rate of differentiating various nonmalignant diseases from AML or ALL is not bad around 90%, this reviewer thinks that this is not a reasonable way to design the study. What is so representative about comparing AML with iron deficiency anaemia or, for that matter, a chronic disease anaemia with an “inflammatory” basis? This reviewer thinks that this is simply a lot of work for no clinical utility at all. Iron deficiency, B12 deficiency and chronic disease anemia can generally be diagnosed by talking to the patient and doing a few simple blood tests. Why would anyone do such a complicated study to distinguish these diseases?? To my way of thinking the weakest part of the study is the comparison with nonmalignant disease. The authors should have chosen other types of non malignant disease which at least have some bone marrow pathology! For instance, some of the hereditary anemias like congenital dyserythropoietic anemia (CDA) which has various forms. Unfortunately, this reviewer had the impression that the authors deliberately put the diagnosis of all the nonmalignant conditions in the Supplementary data since the study does not look so poorly designed if we are not told what exactly these conditions are.
  2. Although the authors say that their goal is to get a type of “gold standard” panel to diagnose residual malignant hematopoietic cells, they used patients’ samples from NEWLY DIAGNOSED patients!!! (line225). This is something that the reader only sees when he/she gets to the Discussion! It is not mentioned in the methods. This point makes the entire study even more unimpressive. The authors can not claim to be building a model of MRD by using exclusively samples of newly diagnosed patients. Well, they can say this is very very preliminary work to see what miRNAs are differentially expressed but really the BEST thing for them to have done would be to forget about iron deficiency and use paired samples of patients of AML and ALL patients before and after chemotherapy, at diagnosis and when they are in remission. This is much more clinically relevant than using iron deficiency anemia or B12 deficiency patients! (by the way it is not clear to me why the patients underwent bone marrow examination for iron deficiency anemia or vitamin B12 deficiency. This reviewer is a hematologist who began practicing medicine in 1980. We did then do bone marrows for diagnosis of iron deficiency anemia since there was no ferritin test clinically available (I am talking about a major teaching hospital in New York City. There was no commercial ferritin test, and the serum B12 test was in its infancy.) Although I do not write that the study is unethical, I hope the authors had a good reason to do a bone marrow on the patients other than to diagnose nutritional anemias.

In case the authors are wondering, this reviewer is also well versed in molecular biology techniques and MRD monitoring of acute leukemias. MRD monitoring is a very important issue for treating acute leukemias, both AML and ALL. At the moment, ALL is usually monitored by specific molecular markers for each patient and AML is difficult to follow at the MRD level unless the patient has a marker that can be tracked like NPM1, PML/RARA etc. FISH can be used for instance trisomy 8 but there are limitations of detection at the MRD level with a range of error that is not optimal at very low levels. The authors in the table of patients’ caharacteristics, mention cytogenetic abnormalities but do not mention if the patients had FLT3 (not a good MRD marker as it is unstable) or NPM1 (a good marker as it is a driver mutation). They do mention what this reviewer would call cytogenetic abnormalities including ones that can be followed at the molecular level such as t8;21. They call these in the Supplementary table “molecular markers” whereas apparently they were diagnosed on the basis of cytogenetics and not RT PCR. The authors could have even correlated the miRNA MRD with the molecular MRD levels for example t8;21 can be monitored by quantitative  RT PCR. Thus the general goal of the study is a good and novel one, however the study design is far from the best one that could have been used.

  1. There is considerable overlap in the box and whisker plots for instance Figure 1. This raises even more questions since individual cases may be totally indistinguishable but we do not see the individual cases. Maybe the authors could put some supplementary figures for a particular miRNA showing each patient’s expression so we can see how much overlap there really is. Basically their statistical significance is highly based on statistical methods whereas the data itself does not appear to be strikingly different among the 3 patient groups (ALL, AML and nonmalignant conditions).
  2. Technical issues with the writing of the paper:
  3. There are a number of places where the authors use background information which is totally wrong. Or perhaps they are translating it from sources incorrectly. For instance the introduction: They say leukemic cells “retain their ability to differentiate into blasts” this is not really correct as blasts are not what we call differentiated. They also use the word “progression in line 47 an 74 and this is not really the correct use of the term progression. Lines 62-63 do not make any sense to me at all. Lines 71-72 it is not clear if they are saying that leukemia cells are heterogeneous within each patient’s disease or heterogeneous among different patients. Lines 84-85 statement about little doubt of the correlation of miRNA with efficacy of chemotherapy needs a reference.
  4. There are a number of other places where the English usage of words is quite incorrect, for example line 265 they say “preventative chemotherapy” which is not something which we give for acute leukemia.

In summary, this reviewer thinks that the optimum thing would be to redo the study with more appropriate patients, otherwise it needs to have a very major rewrite to explain their goal and findings in a more realistic way.

Round 2

Reviewer 2 Report

The authors have revised their manuscript and it is definitely better.

However there are still two points (one very minor and one very major) that need clarification.

  1. The minor point is that in the Introduction, they changed the sentence I pointed out in which they describe leukemia cells differentiating into blasts. However apparently they did not really read a lot about acute leukemia since they still left the sentence that there is PANCYTOPENIA in acute leukemia. The authors are correct in saying that the leukemic cells crowd out the normal precursors of the marrow but they incorrect in the way that they say pancytopenia is the presentation of acute leukemia. Most of the time there is leukocytosis with associated anemia and thrombocytopenia. Pancytopenia is present in a minority of cases.
  2. The main problem (which most of my previous review dealt with) is still present. The authors have left the definition of the NTP cases (nontumor pathology) in the Supplementary data and this is misleading. If they are so convinced that these patients (with iron deficiency or anemia of chronic disease, B12 deficiency etc) are a GOOD COMPARATOR for malignant miRNA levels, well, they should OPENLY STATE THIS IN THE TEXT instead of hiding it in the supplementary data. Personally I am not so convinced that this is true. But it can be construed as a legitimate attempt to establish a methodology. Supplementary Figure 1 shows a certain degree of overlap and I also do not understand why the authors put this in Supplementary data when they left an endlessly long table of the various miRNAs in the body of the main text. It is visually more effective to see Supplementary Figure 1 than to plow through that table. In conclusion this reviewer personally thinks that the manuscript should not be accepted unless the authors are more honest and forthright. I think that the experiment was legitimate but the reader (I assume the authors would like to have people read this study?) has to understand what the NONTUMOR PATHOLOGY cases included otherwise he can not figure out what is the TRUE MEANING of the results presented!

In conclusion, I feel that, if the authors think they did the right comparison, they should stand by their data and make it clear what they did, instead of trying to hide this crucial information about the experimental design by putting it in Supplementary data which not all readers bother to go into.

Round 3

Reviewer 2 Report

This reviewer thanks the authors for their revisions. They have added a few sentences which are actually very appropriate in explaining the reason for the choices of patients whose samples they chose as comparators for the leukemias and this is a completely logical explanation. Furthermore they have moved supplementary table 1 to the text and not as part of supplementary information so as to clarify the study design. They also moved a scattergram to the body of the text instead of being in Supplementary figures and they changed the box and whisker diagram and the scattergram to colored figures so it is much easier visually to see what is going on.

All of these changes have made the manuscript more comprehensible and more easy to read.

I have no further suggestions for revisions.